# Design a Clinical Research Protocol: Influence of Real-World Setting

**DOI:** 10.3390/healthcare11162254

**Published:** 2023-08-10

**Authors:** Jonathan Cimino, Claude Braun

**Affiliations:** 1Clinical Research Unit, Fondation Hôpitaux Robert Schuman, 44 Rue d’Anvers, 1130 Luxembourg, Luxembourg; claude.braun@hopitauxschuman.lu; 2Hôpitaux Robert Schuman, 9 Rue Edward Steichen, 2540 Luxembourg, Luxembourg

**Keywords:** clinical research, ethics, research methodology regulatory

## Abstract

The design of a clinical research protocol to evaluate new therapies, devices, patient quality of life, and medical practices from scratch is probably one of the greatest challenges for the majority of novice researchers. This is especially true since a high-quality methodology is required to achieve success and effectiveness in academic and hospital research centers. This review discusses the concrete steps and necessary guidelines needed to create and structure a research protocol. Along with the methodology, some administrative challenges (ethics, regulatory and people-management barriers) and possible time-saving recommendations (standardized procedures, collaborative training, and centralization) are discussed.

## 1. Introduction

The Accelerating Clinical Trials (ACT) in the Europe (EU) initiative aims to develop the EU further as a competitive center for innovative clinical research with the creation of national centers and/or institutions, like EU medicines agencies, dedicated to health research and to delivering on the clinical-trial-innovation recommendations of the network strategy. According to the European Medicines Agency (EMA), approximately 4000 research projects are authorized each year across the European Union (EU). These studies are conducted under the sponsorship of various organizations, which take responsibility for managing, funding, monitoring, collecting, and analyzing data. The pharmaceutical industry acts as the primary sponsor for 60% of these projects, while non-commercial sponsors or academic institutions account for the remaining 40%. Ultimately, these studies have a positive impact on the life expectancy and quality of participants in the EU [1]. According to the guidelines set by the International Council for Harmonization of Technical Requirements for Pharmaceuticals for Human Use (ICH) Guideline for Good Clinical Practice (GCP) (ICH-GCP), a clinical trial refers to any research conducted on human participants involving the use of an investigational medicinal product (IMP). This product can be a pharmaceutical substance or a placebo, and the trial typically spans four phases, 1 to 4. Significant factors to consider are the inclusion of unlicensed therapies [2,3,4] or the utilization of drugs beyond their approved usage [5], such as for novel indications. Clinical trials play a crucial role in establishing the safety, appropriate dosages, and effectiveness of new treatments for diseases. They also contribute to the development of innovative methods for disease detection, diagnosis, and prevention. Clinical-trial investigations represent the lengthiest and costliest stages of drug development, with an average timeframe of 15 years until gaining approval for patient use [6,7,8,9]. Fortunately, European harmonization tools, like the Clinical Trials Information System (CTIS), are now in place to facilitate regulatory and multicenter submission. The Clinical Trial Information System (CTIS) facilitates the exchange of information among clinical trial sponsors, EU Member States, European Economic Area (EEA) countries, and the European Commission. From 31 January 2023 [10], the utilization of CTIS became obligatory for new clinical-trial applications in the EU. However, it is important to note that the research endeavor encompasses more than clinical trials alone. It also encompasses a wide range of studies, including observational research, cost-effectiveness analyses, quality-of-life assessments, translational studies, and competitive machine-learning investigations [11,12]. Research studies can be conducted in various settings, such as academic institutions, community clinics, or the private offices of doctors. However, hospitals hold a unique advantage in identifying emerging clinical areas of interest and providing patients with access to therapies that are still undergoing regulatory processes or that are limited by cost or health-insurance restrictions, making them unavailable on the market for general patient use [13].

Clinical research must be conducted according to a protocol. Reference guidelines include the Standard Protocol Items: Recommendations for Interventional Trials (SPIRIT) statement and ICH-GCP, and reporting guidelines, like the Consolidated Standards of Reporting Trials (CONSORT). According to the ICH E6 (R2) guidelines, a clinical-trial protocol serves as a comprehensive document outlining the objectives, design, methodology, statistical considerations, and overall organization of a study. It provides a detailed description of each critical aspect of the study, elucidating the actions and procedures to be undertaken. The protocol encompasses the scientific background, study objectives, primary and secondary endpoints, target population, activities at each study visit, minimum participant enrollment required for statistical power, and the data-analysis plan [14,15,16]. Additionally, the protocol includes information regarding previous study phases, participant eligibility and exclusion criteria, a schedule of tests and procedures, risk–benefit assessments, data-collection methods, and the duration of the study [17]. The protocol must be validated by competent authorities (CAs) and ethics committees (ECs), balancing clinical effectiveness with research design [18].

Research protocols must provide a clear and complete design flow to meet study objectives. In order to write a clear and accurate protocol, principal investigators must have a thorough understanding of the trial design and be willing to review and discuss its various aspects with protocol-development teams. However, several challenges can impede the translation of ideas into successful research projects. These challenges include a lack of research knowledge, difficulty in executing plans, a lack of familiarity with the research landscape, and a multitude of regulations. Furthermore, engagement in research is a laborious undertaking that frequently requires that researchers navigate intricate administrative procedures [19]. These various factors can create substantial obstacles to the performance of research of exceptional quality, a crucial aspect in advancing scientific understanding and enhancing healthcare outcomes. Consequently, without a sound methodology, it is unreasonable to expect a clinical research venture to succeed. However, by capitalizing on institutional resources, such as researchers, statisticians, and methodologists, the establishment of a standardized research protocol can effectively surmount these barriers and prioritize clinical research within healthcare institutions [20,21].

This article primarily aims to provide a comprehensive guide to designing a research protocol, offering a step-by-step approach. Additionally, it reviews prominent standardized clinical-research guidelines that new researchers should expect to encounter while formulating their protocols.

## 2. Methodology

To ensure the success of any clinical protocol, it is crucial to have the appropriate “building blocks” in place. These “building blocks” encompass essential components, such as scientific methodology, people-management skills, ethics and regulatory compliance, financial dynamics, participant recruitment, information technology and systems, and institutional commitment. The effective organization and operational communication of these elements, including activities like financing, patient recruitment, informed-consent processes, safety and deviation from reporting in line with the protocol, administration or destruction of investigational medicinal products, and staff training, are vital for designing clinical research across various phases, ranging from observational to investigational. It is imperative that this entire process adheres to the comprehensive guidelines outlined by EQUATOR [22]. The step-by-step creation of a research protocol for interested young researchers is described in this review.

### 2.1. Building Blocks of Clinical-Research Protocol

As with the inception of any novel clinical-research project, a protocol should begin with a solid foundation and gradually expand. The process involves eight essential blocks that progress incrementally, starting from the conceptual stage prior to drafting, advancing to the initial drafting phase, moving towards submission and regulatory validation, and ultimately culminating in the publication of the protocol. Each of these building blocks plays a vital role in fostering the growth and durability of the protocol’s foundation. In Figure 1, essential resource elements are depicted, highlighting their significance and the desired outcomes, which are crucial for the development and success of a clinical research protocol. Certain blocks are interdependent, meaning that factors such as available funding and its projected trajectory affect various aspects of the study, including its duration, required examinations and biological analyses, infrastructure growth, and the recruitment of human resources. Furthermore, the integration of SMART and FINER criteria, along with a robust strategic approach, enhances the effectiveness of the methodology and the statistical analysis of both primary and secondary outcomes [23]. Moreover, it is crucial to acknowledge that the blocks involved in the process can be subject to legal obligations, such as EU regulations, which establish specific criteria regarding IT infrastructure, storage, and data protection [24,25,26,27,28].

### 2.2. Before Writing Phase: Feasibility, Study Aims, and Methodology

The feasibility phase is clearly the most important for any clinical researcher, and, unfortunately, it is the most inconsistent [29,30]. One of the first questions that needs to be answered before writing a protocol can be summarized as:“What are the aims of the study?”, This encompasses the SMART (specific, measurable, achievable, relevant, and time-frame) criteria.“Why, Where and How the study should be conducted?”. This encompasses the FINER (feasible, interesting, novel, ethical, and relevant) criteria.

When preparing a proposal, it is vital to focus on essential aspects, which comprise the provision of a solid rationale for the project’s necessity and the outline of a comprehensive investigative plan. It is advisable to thoroughly comprehend the requirements prior to commencing the writing process in order to ensure accuracy and relevance. These questions are therefore linked to two aspects: the pure research question (clear design, objectives, endpoints, procedures, and methods to be used) and the surrounding environment (operational, scientific, and financial resources required). Some crucial points to consider in order to raise the research question and clarify its importance are presented below (Figure 2).

These points include general study objective, which reflects all scientific questions (especially limitations), purpose and scope of question to be answered by the trial, the primary (for which the trial is powered) and secondary endpoints, and time frame. A key component of a clinical trial is the selection of appropriate endpoints, which are the outcomes used to measure the treatment’s effectiveness. Endpoints can be grouped into several major types, including clinical endpoints, surrogate endpoints, patient-reported outcomes, and biomarkers [31,32,33,34,35,36,37,38,39,40,41,42,43,44,45]. Clinical endpoints are the most direct measures of a treatment’s impact on patient health, including mortality, disease progression, or symptom relief [31,32,33,34]. Surrogate endpoints, on the other hand, are indirect measures of treatment effectiveness, such as blood pressure or cholesterol levels, which are used as proxies for clinical outcomes [35,36,37,38,39,40,41]. Patient-reported outcomes are self-reported measures of how patients feel or function, such as quality-of-life or pain scores [42,43,44,45]. Biomarkers are physiological or molecular measures that are used to assess disease status or treatment response [46,47,48,49,50]. The selection of appropriate endpoints is critical to the success of a clinical trial, as it can influence the trial’s design, sample size, and statistical power. However, there are several challenges associated with endpoint selection, including defining and validating the endpoint, ensuring its relevance to patients, and minimizing bias in its measurement. In recent years, there has been a growing recognition of the importance of patient-centered endpoints, which prioritize outcomes that are meaningful to patients, such as improved function or quality of life. The use of patient-centered endpoints can improve patient engagement, satisfaction, and adherence to treatment, as well as providing valuable insights into the patient experience [51,52]. An important part of this process is the study design (Figure 3). Clinical research can attempt to answer various questions about drugs, medical devices, patient questionnaires (patient-reported outcomes or experience measures), and analyze collections of biopsies, etc. Various study designs are specifically applied in the pursuit of these goals.

Essentially, there are two major types of clinical-study design: non-interventional (observational studies, sometimes called epidemiologic studies) and interventional (experimental). Each study design has its own inherent weaknesses and strengths. The selection of a study design depends on various factors, as described in the following discussion. These factors include prior research goals and results, the availability of retrospective data or study participants, funding and resource availability, and time limitations [53]. Observational studies serve as hypothesis-generating investigations and can be categorized into descriptive and analytic subtypes. Descriptive observational studies aim to provide an overview of the exposure and/or outcome under investigation, while analytic observational studies focus on measuring the association between the exposure and the outcome. Observational studies (Figure 3) include many subtypes (these are interesting, fast, and very inexpensive, but sometimes more limited; they include meta-analyses, systematic reviews, case reports, case series, and ecological studies). The most common designs include:(i)Case–control studies, which are conducted retrospectively and are relatively straightforward to execute. They involve comparison between two distinct groups: a group with a disease (cases) and a group without this disease (controls). For instance, researchers might examine the likelihood of individuals being diabetic among obese patients compared to non-obese individuals. To accomplish this, a cohort of patients diagnosed with diabetes is selected, along with a control group exhibiting normal blood-sugar levels. Their weight histories are then examined to identify any potential correlations [54,55].(ii)Cohort studies employ a longitudinal design, which can be retrospective, using medical records, or prospective, by following participants over time. These studies involve comparisons of two samples from a population, one with a specific risk factor and the other without this risk factor, by measuring the relative risk (RR). For example, researchers might investigate the RR of developing non-small-cell lung cancer (NSCLC) among people who smoke. A sample is selected, consisting of both smokers and non-smokers, and the number of individuals with NSCLC is calculated within each group [56].(iii)Cross-sectional studies (translational design, easy to conduct) are observational research designs that aim to collect data on a particular population at a specific point in time. Unlike longitudinal studies, which follow a group of individuals over an extended period, cross-sectional studies provide a snapshot of the population’s characteristics, behaviors, or attitudes at a given moment. This type of study can be conducted using surveys, interviews, or physical examinations to gather information on various variables of interest. Cross-sectional studies are often used to explore the prevalence of certain diseases or risk factors in a population, as well as to identify associations between variables. However, it is important to note that cross-sectional studies do not establish causality and are limited by their inability to capture changes over time. As an example, the examination of the prevalence of smoking among teenagers in a particular city can be considered. The researcher selects a sample of teenagers from different schools in the city and administers a survey that asks them about their smoking habits. The survey also collects demographic information, such as age, gender, and socioeconomic status. The researcher then analyzes the data to determine the proportion of teenagers who smoke, as well as any patterns or associations between smoking and demographic factors [57,58].

In contrast, experimental studies (Figure 3), by definition, investigate the effect of a specific intervention (e.g., therapeutic agents or prevention) and the association between the exposure and the outcome. In other words, the risk factor/exposure of interest/treatment is controlled by the investigator. The gold standard and strongest interventional study designs are randomized controlled trials (RCT), prospective studies that measure the effectiveness of a new intervention or treatment against an existing standard of care (or control treatments, or placebo). This design uses randomization (which reduces bias to examine cause–effect relationships) [59,60]. It is crucial to remember that every study design varies, making it essential to select a design that can effectively address the specific question at hand and yield the most valuable insights. Consequently, meticulous planning and allocation of resources, both human and financial, are necessary to ensure that a study’s design and scientific validations are conducted with precision, ultimately leading to accurate results.

After reviewing critical points on research questions, we discuss the second limiting point in the design of clinical research: the role of the environment surrounding the researcher [61]. In terms of infrastructure (hospital, clinical-research centers, etc.), a limitation that investigators must check is the budget that is allocated to the study (Figure 4). A significant proportion of phase 3 studies fail due to a lack of funding [62,63]. The financial management of research activities is complex and budgets may be drawn from multiple sources, such as pharmaceutical industry (company sponsored trial), university–EU–Foundation funds (academic-investigator-sponsored trials). The budgets are mainly planned to cover the costs of human-resource management in the study (investigator teams, contract, screening and inclusion of patients, etc.), interventional procedures (treatments, exams, laboratory analyses, etc.), data management (databases, CRF design, statistical software, CRF filling, support during monitoring, etc.), and administration (specific insurance, stipends for included patients, institution-review fees, etc.).

In general, the fees associated with a study can vary based on whether it is commercial or academic in nature. Commercial studies sponsored by pharmaceutical companies typically have larger funding allocations, whereas academic consortia often operate with more limited funding. However, academic studies offer the advantage of involving investigators who can contribute to scientific publications. Ultimately, the determination of fees is negotiated by parties involved. Regardless of the study type, it is crucial to recognize that financial investment should never be underestimated. The only viable approach is to anticipate project expenses accordingly, as emphasized in previous research [64,65].

In terms of infrastructure (hospitals, clinical-research centers, etc.), the second point must be focused on direct and indirect collaborators. In many instances, investigators who have access to robust research infrastructure can enhance their knowledge development, enhance healthcare delivery, and seamlessly incorporate these aspects into clinical practice [61]. Table 1 illustrates the composition of an ideal internal research team. Clinical research units strive to alleviate the challenges associated with investigator involvement in clinical trials, along with supporting departments such as pharmacy, IT, and laboratory personnel.

Although the composition of clinical research teams may vary, certain key roles should be filled to ensure that the team functions effectively. The ideal team structure consists of a proficient medical director responsible for research and medical affairs, competent experts (including nurses, research associates and coordinator, data managers, etc.) capable of handling clinical and administrative tasks, scheduling, and data management, and capable finance/legal/business personnel to oversee contracts, budgets, and business strategies that enhance site growth. It is crucial to have team members with established networks and relationships to attract investigational studies. Job descriptions should prioritize consistency and transparency in job performance [66]. The clinical staff’s responsibilities encompass various tasks, including screening potential study participants, determining eligibility, coordinating patient schedules, preparing regulatory- and ethics-committee submissions, managing amendments, reporting safety data, providing patient education, obtaining informed consent, and evaluating potential adverse events. The team’s roles are influenced by the size and business model of the institution. In smaller institutions, a study coordinator may undertake multiple duties, such as contracts, regulatory affairs, and recruitment. Larger research sites typically employ separate personnel for clinical, data-management, legal, and administrative roles. Additionally, they have a dedicated business-development team to foster relationships. Close collaboration with statisticians, data protection officers (DPOs), and quality officers at the institution is essential. Training and continuous assessment of knowledge are crucial for all key players, including pharmacy and laboratory staff, to ensure long-term sustainability. Furthermore, an experienced medical director overseeing research and medical affairs, along with individuals possessing the necessary network and relationships to attract investigational studies, are vital. Lastly, skilled finance/legal/business personnel are necessary for managing contracts and budgets and implementing effective business strategies to optimize site growth. One way to ensure this is the certification of research staff in good clinical practices, with regular refreshers every two years. In this way, regardless of size, all research teams (which are generally employed by research centers) perform similar roles and, in most cases, small number of employees carry out several of these activities. The PI must ensure that each of the functions (sometimes involving particular areas of expertise, such as IT, statistical analysis, or randomization coordination) is carried out with the required skills (and registered in the delegation log of the study).

A perfect example that brings together all the points above (methodology, budget, availability, and the competence of the team) is the following question: Is there a statistician available to calculate the minimum number of patients needed to reach the primary endpoint of my study? Statisticians play a critical role in research studies. They calculate the sample size and power, establish objectives and endpoints, and consider the detection of clinically significant differences between treatment groups. Furthermore, sample sizes are crucial for meaningful results. In addition, statisticians also ensure that study designs minimize variability and determine the need for interim analyses.

### 2.3. Writing Phase: Guidelines and Observed Discrepancies

Protocol writing is a step that enables researchers to assess existing knowledge, conduct a comprehensive review of relevant research (including preclinical and clinical data, both published and unpublished), and critically evaluate the topic of interest. This involves planning and reviewing project steps, providing a roadmap for the entire investigation, and serving as a valuable guide throughout the research process.

Submitting a protocol that lacks critical details is a regulatory risk. This can be completely avoided by referring to a checklist of what should be included in a clinical study protocol, like SPIRIT and CONSORT for reporting, which are evidence-based guidelines [67,68]. These statements offer authors guidance on the essential information that must be included when reporting trials. Their purpose is to ensure that both protocols and related reports are characterized by clarity, comprehensiveness, and transparency. Different protocol templates are available, like ICH M11 template for EU, FDA (Food and Drug Administration) or NIH (National Institutes of Health) template for US, Health Research Authority (NHS) for UK, etc. Typically, the contents of a clinical research protocol include the following topics (Table 2).

For example, although this may seem basic, it is important for researchers to complete the format of their protocol in a way that allows readers to efficiently navigate into specific sections (page numbers, section headings, etc.). This increases productivity and eliminates any unnecessary confusion for clinical-site staff and regulatory/ethics authorities. The same applies to the title of the study, which must be accurate, short, concise, and identifiable.

Therefore, this first part is mainly linked to the format, the required items, and the understanding of the study plan by the reader. The second part corresponds to the search for the most common discrepancies in order to anticipate failure. This involves the ability to anticipate and increase the number of patients included, to highlight the effectiveness of a treatment compared to another, to delete therapeutic arms as results are obtained, etc. [69]. Inflexible conventional clinical trials have been run in three steps, without the inclusion of options for changes during the course of trials: (i) the trial is designed; (ii) the trial is conducted as prescribed by the design;(iii) the data are analyzed according to a pre-specified analysis plan (Figure 5A). The choice of adaptive designs [70,71] (which can be applied across all phases of clinical research) can make clinical trials more flexible by utilizing results before an interim analysis (sometimes several) to modify the trial’s course (like sample-size re-estimation to ensure adequate power is maintained) (Figure 5B).

The most common adaptive trials are:(i)Sample-size reassessment (Figure 5(B1)): sample size can be reassessed (after interim analysis) and increased to ensure that the trial is adequately powered.(ii)Response adaptive randomization (Figure 5(B2)): treatment-allocation ratio can be modified (after interim analysis) to favor enrolment in specific treatment.(iii)Adaptive enrichment (Figure 5(B3)): study-eligibility criteria can be modified with a sample-size reassessment (after interim analysis) to investigate the efficacy of the intervention in the related subgroup.

Finally, another type of adaptive trial, which allows several treatments to be assessed concurrently with or separately from preplanned interim adaptations, is the so-called multi-arm multi-stage (MAMS) design (Figure 5C). In recent years, the use of adaptive clinical trials has gained momentum, and several major adaptive clinical trials have been conducted worldwide [70,71]. One example is the REMAP-CAP trial, a multi-center, randomized, adaptive-platform trial for community-acquired pneumonia, which has been adapted to investigate treatments for COVID-19 [72]. Another example is the I-SPY 2 trial, a platform trial for breast cancer that uses adaptive randomization to test multiple investigational drugs simultaneously [73]. These trials are reshaping the way in which clinical research is conducted, and their success has led to increased interest and investment in this field. However, there are also challenges associated with adaptive trials, such as statistical and regulatory issues, which need to be addressed.

Failures can stem from various sources, including the absence of a well-defined strategy to achieve objectives, challenges related to study duration and compliance, inadequacies in the administration of the investigational drug, insufficient data to conclude a trial, and other factors, such as failure to adhere to inclusion criteria, disregard of EU guidance, or difficulties in patient recruitment, enrollment, and retention. It is crucial to generate precise and substantial results at every stage of the clinical-trial process in order to make informed decisions about whether to continue the study. This is why protocols should always be subjected to rigorous risk assessment before their submission to the authorities. Whenever possible, it is strongly recommended to involve experienced people in a given therapeutic area, like in the use of external clinicians, key leader opinions (KOLs), and academic and private partners, like contract research organizations (CROs).

### 2.4. Internal Review

The internal review of a protocol before regulatory submission is an essential phase, and all contributors are encouraged (in a team effort) to discuss the implications of the study’s planning for current clinical practice. Unfortunately, this is a frequently neglected step. The aim of a clinical protocol is to provide a concise plan outlining the rationale for and design of a planned study. The purpose of the internal review (involving the different profiles/backgrounds of the team, see Table 1) is to force investigator teams to clarify their thoughts and to consider all the aspects of the study. For example, this can be achieved by following the progress of a patient in the study (from their recruitment/inclusion, with consent, until their last visit), through the potential randomization, allocation of treatment, clinical examination procedures, encoding of the data, to the potential role of the pharmacy and the laboratory, etc.

The PI is a key contributor to the review of the protocol, re-determining the selection criteria, the design of the study, its objectives and endpoints, and its dose escalation, the revaluation of the safety data, changes in concomitant therapies (disallowed or permitted), etc. While conducting an internal review of a protocol, investigators should seek to capitalize on the insights and expertise of colleagues and experts to enhance and fine-tune their plans. The backbone of the protocol must be complete, clear, and, above all, practical, before regulatory submission [74,75].

### 2.5. Regulatory and Ethics Validations

An institutional review board (IRB) is a committee that is responsible for review and approval of research studies involving human participants. The primary function of the IRB is to ensure that studies meet ethical and regulatory standards and that the rights and welfare of study participants are protected. The IRB is typically composed of a diverse group of individuals, including scientists, healthcare professionals, ethicists, and community members. Before a research study can begin, it must undergo review by an IRB. The IRB evaluates the study’s protocol to ensure that it meets ethical and regulatory guidelines, including obtaining informed consent from participants, minimizing risks to participants, and protecting participant confidentiality. The IRB also monitors ongoing studies to ensure that they continue to meet these standards and may request modifications to or terminate a study if necessary. Both the Declaration of Helsinki and the ICH-GCP guidelines were developed to provide a unified standard for the design, conduct, and reporting of clinical trials involving human subjects, including study design, ethical considerations, participant safety, and data management [76,77].

The new Clinical Trial Regulation (CTR) 536/2014 is a regulatory framework that governs the conduct of clinical trials in the European Union (EU). The CTR, which replaced the previous Clinical Trials Directive, aims to streamline the regulatory process for clinical trials and harmonize the rules across the EU member states. The CTR introduces a centralized EU portal and database for the submission and assessment of clinical-trial applications, making the application process more efficient and consistent across the EU. It also introduces new requirements for informed consent, transparency, and the public disclosure of clinical-trial results. The CTR places a strong emphasis on the protection of study participants, including vulnerable populations, such as children and people with disabilities. It requires that clinical trials are conducted in accordance with the principles of good clinical practice (GCP) and that study sponsors have appropriate procedures in place for the management of adverse events and the protection of participant data. Overall, the CTR represents a significant step forward in the regulation of clinical trials in the EU. It is designed to improve the efficiency of the clinical-trial process while ensuring the highest standards of participant safety and data integrity. The CTR is set to come into effect in 2022, and it will apply to all clinical trials conducted within the EU, as well as to trials conducted outside the EU that involve EU participants or data [78]. Under the new Clinical Trial Regulation (CTR) 536/2014, the sponsors of clinical trials must fulfill several obligations to ensure the safety and welfare of study participants and the integrity of study data. Some of the key obligations of sponsors include:A.Submitting a clinical-trial application to the European Medicines Agency (EMA) and obtaining a favorable opinion from the Member State Ethics Committee before initiating a trial.B.Designing clinical trials in accordance with good clinical practice (GCP) guidelines and ensuring that studies are scientifically valid, feasible, and ethical.C.Ensuring that all the study staff involved in a trial are appropriately qualified and trained to carry out their duties.D.Establishing procedures for the monitoring, recording, and reporting of adverse events and serious adverse events that occur during trials.E.Ensuring that informed consent is obtained from all of a study’s participants, and that they are provided with clear and accurate information about the trial.F.Ensuring that trials are registered on a publicly accessible database, and that the results of trials are publicly disclosed.G.Ensuring that trials are conducted in compliance with applicable data-protection and privacy laws.

At the institution level, standard operating procedures (SOPs) are critical documents that provide detailed instructions on how to perform specific tasks or activities in a consistent and standardized manner. The SOPs for clinical trials cover a wide range of activities, including study design, participant recruitment, informed consent, data collection and management, safety monitoring, and adverse-event reporting. These procedures are developed by the sponsor or research organization conducting the trial, and they are based on established regulatory requirements and good clinical practice (GCP) guidelines. The purpose of SOPs in clinical trials is to ensure that all study personnel are trained and knowledgeable in the procedures required for the conduct of trials. They provide a clear and consistent framework for the execution of study activities, as well as helping to ensure that trials are conducted with integrity and that the safety and welfare of study participants are protected. Furthermore, SOPs play a critical role in ensuring that trials are conducted in compliance with applicable regulatory requirements, and that the data generated from trials are accurate, reliable, and verifiable. They are regularly reviewed and updated throughout trials to ensure that they remain current and relevant. The PI must submit all the request documents (with additional documents for interventional-drug and medical-device trials) to the CA and EC at national or international levels, depending the countries involved (Figure 6).

### 2.6. Registration of the Protocol

The protocol registration is the reference for notifying international scientists (public database) that a related clinical study is underway around the world. Not only does this help to avoid duplication, as well as saving time and resources, but it also paves the way for collaborative work among researchers with common interests in specific fields. The PI (or their representative) bears the responsibility of ensuring timely study registration, reporting, and updates, according to the timelines set by relevant regulatory bodies for both new and ongoing clinical trials. These bodies may include platforms such as ClinicalTrials.gov, the WHO registry, EudraCT, and others. For instance, consider the case of ClinicalTrials.gov, where the PI is accountable for fulfilling the following functions [79,80]:The selection of which trial to register on the publicly available database;Requests for users to set up accounts (PI organizations, like hospitals, universities, medical centers, etc.);The creation ClinicalTrials.gov research contents;The validation of the accuracy of entered content;Checking “Record Status” on ClinicalTrials.gov;The validation of notification emails from ClinicalTrials.gov within 15 (registration) and 25 days (results);Maintaining records entered on ClinicalTrials.gov in line with policy lines;Contacting ClinicalTrials.gov in case of expected/unexpected PI changes within 15 to 30 days.

The records on ClinicalTrials.gov comprise three sections: protocols, results, and documents. The protocol section serves as the initial registration record, containing the study title, NCT number, and document date. For studies on ClinicalTrials.gov, the minimum registration requirement is the protocol section, which must be reported no later than 12 months. Additionally, the authors of these studies are required to add and complete results records and upload protocols in PDF/A format.

## 3. Discussion

Clinical trials are essential in drug development, but they can be complex and challenging to design and execute. To ensure the success of a clinical trial, it is crucial to establish a well-designed protocol that answers the research question adequately. The protocol must be sufficiently strong to produce the expected knowledge, feature a statistically and ethically justified number of participants, and provide accurate details about the methodology to allow reproducibility by other investigators. However, field practitioners often struggle to anticipate issues that may arise during the trial. The failure to demonstrate efficacy and ensure safety remains the primary source of interventional trial failure [81,82]. There are various reasons why potentially effective drugs may not demonstrate efficacy, such as an inappropriate study design or statistical endpoint, or an underpowered trial, leading to patient dropouts.

Another critical point the insufficient consideration of the utilization of real-world evidence (RWE)-based clinical studies in protocol design. These RWE clinical studies have emerged as valuable approaches in medical research, offering several benefits and challenges. These studies provide enhanced external validity by reflecting real-world patient populations more accurately than traditional randomized controlled trials (RCTs). The RWE studies encompass broader spectra of patients, leading to broader generalizability and informed clinical decision-making based on individual patient needs. Additionally, RWE studies offer longitudinal insights by observing patients over extended periods, revealing long-term treatment outcomes and safety profiles. One of the significant advantages of RWE studies is their cost-effectiveness. The leveraging of existing data sources, like electronic health records and claims databases, reduces the need for extensive infrastructure and patient recruitment, expediting the evidence-generation process. Ethically, RWE studies address certain concerns by observing treatment effects without intervening in patient care, avoiding the need for placebo-controlled arms in some cases. However, RWE studies also face several challenges. Ensuring data quality and completeness is paramount, as electronic health records and claims databases may contain missing or erroneous information, potentially introducing bias. Selection bias and confounding factors are inherent risks in RWE studies, as patients may self-select treatments, leading to imbalances in patient characteristics across treatment groups. Statistical techniques, like propensity-score matching and sensitivity analyses, are employed to mitigate these biases, but their effectiveness relies on the quality and breadth of the available data. The lack of standardization in RWE studies presents another challenge, as protocols and data-collection practices can vary significantly across studies, making direct comparisons difficult and hindering generalizability. Data-privacy and security concerns arise with the increasing reliance on electronic health records and digital sources, necessitating robust data anonymization and secure data-sharing protocols to protect patient confidentiality and comply with data-protection regulations. The determination of causality in RWE studies is challenging due to their observational nature. Unlike RCTs with randomization and control groups, RWE studies rely on statistical methodologies and causal inference techniques to establish links between treatments and outcomes. While sophisticated analytical approaches exist, they cannot fully account for all potential confounders, leading to inherent uncertainties in causal interpretations [83,84].

Therefore, it is essential to take all these factors into account when designing a clinical trial protocol. Other parameters have to be taken into account. For example, in investigator-sponsored trials, poor training and a lack of collaboration often contribute to these problems. To address this, hospitals should gain a clear understanding of the research occurring at their facilities and of their role. They can then take steps to support clinical investigators and ensure they have the necessary training and resources to conduct clinical trials successfully [85]. Physician motivation and the time dedicated to research can vary, depending on the institution type. Clinical research is often better supported within academic or public institutions, where physicians and their staff members may receive salaries that specifically fund clinical-research activities [86]. This arrangement offers valuable employment opportunities for individuals committed to conducting trials, relieving them of the pressure to solely focus on increasing patient volumes. As a result, they can allocate additional time and effort to dedicated clinical-research endeavors. Moreover, various organizations offer support to clinical investigators. For example, the European Clinical Research Infrastructure Network (ECRIN) provides guidelines, protocol templates, educational programs, and protocol-writing conferences to support young clinical investigators [87]. In addition, the most recent medical schools include courses in research design to prepare future clinical investigators.

Although these efforts may seem insufficient, they can provide crucial support to overcome the challenges of conducting clinical trials in real-world settings. Overall, the design of an effective clinical trial protocol requires the careful consideration of many factors and collaboration among various stakeholders to ensure success.

## 4. Conclusions

The protocol is the core of clinical research, which every motivated clinician attempts to perform. It is a scientific road map that aims to justify, clarify, and highlight the research questions, overall methods, design, and analytical approaches. Above all, the writing of a protocol is a team effort. Any scholar aiming to conduct research to help patients should start by developing, writing, and disseminating a study protocol. Well-written clinical protocols facilitate the successful conduct of trials. Clear and consistent writing with early feedback also ensures that studies are feasible, practical, and cost-effective, as well as minimizing the potential for avoidable amendments. This review provides a checklist of the essential minimum requirements for developing a precise clinical-research protocol, offering valuable guidance in the process.

## Figures and Tables

**Figure 1 healthcare-11-02254-f001:**
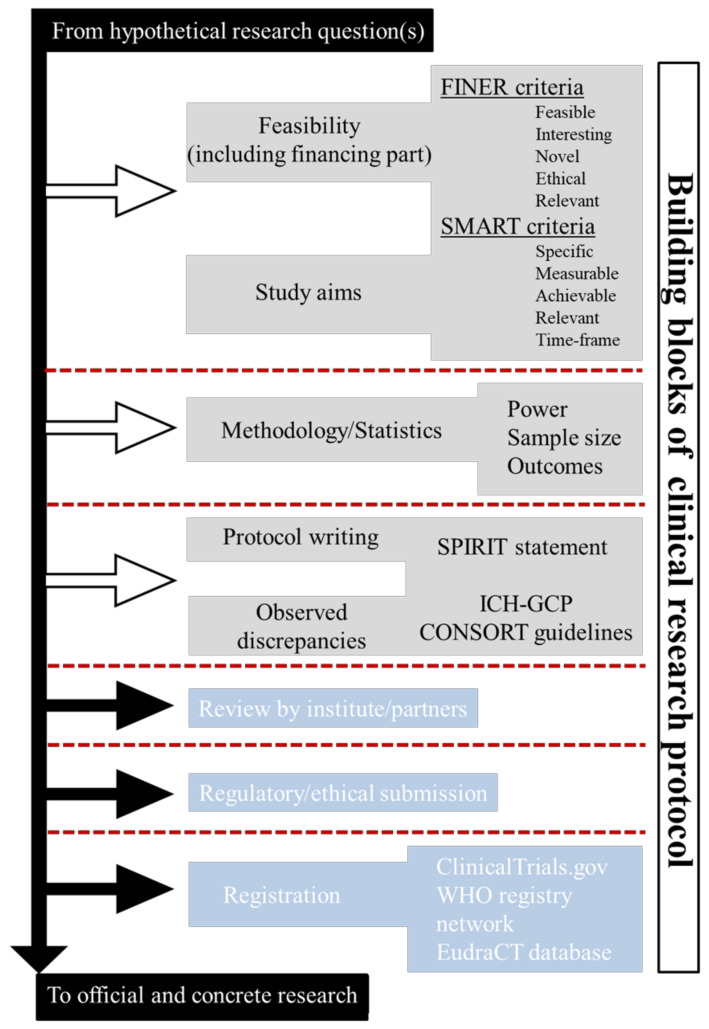
Foundational elements and desired outcomes essential for the progress and success of a clinical-research endeavor.

**Figure 2 healthcare-11-02254-f002:**
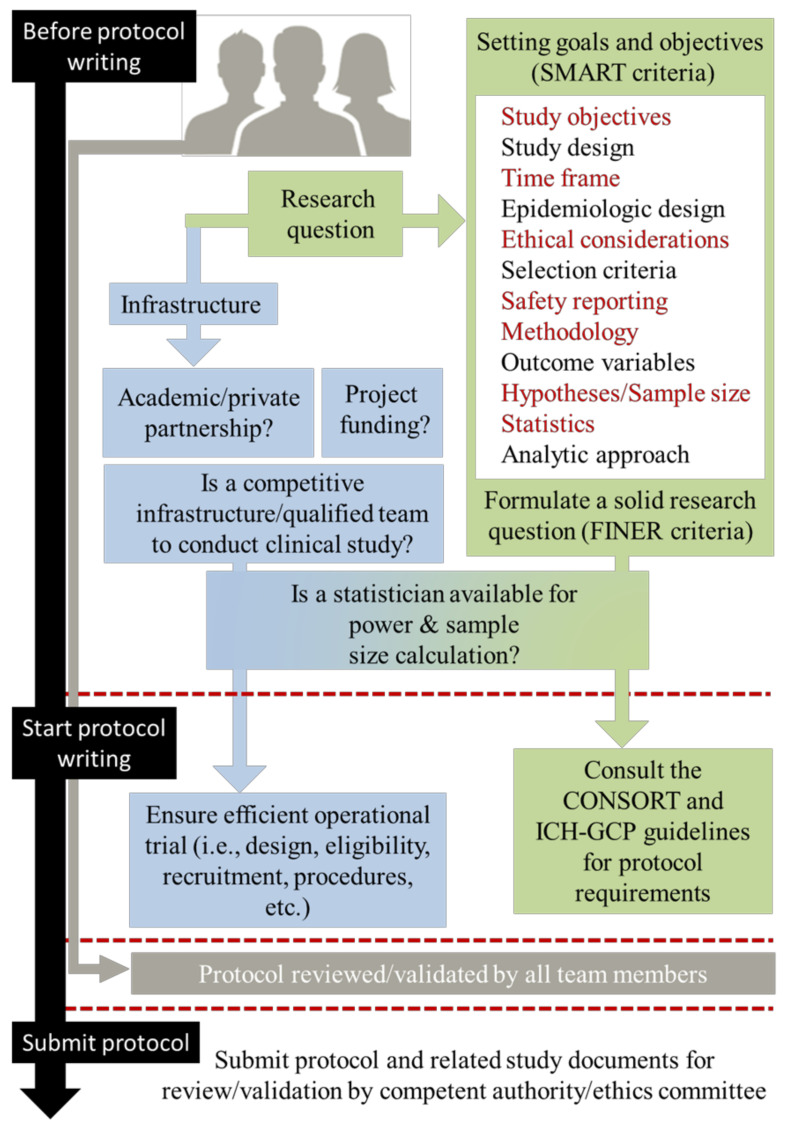
Protocol design: Before–during–after writing phase (feasibility, study aims, methodology, guidelines and observed discrepancies, internal and external review, regulatory and ethics validations, final registration of the protocol).

**Figure 3 healthcare-11-02254-f003:**
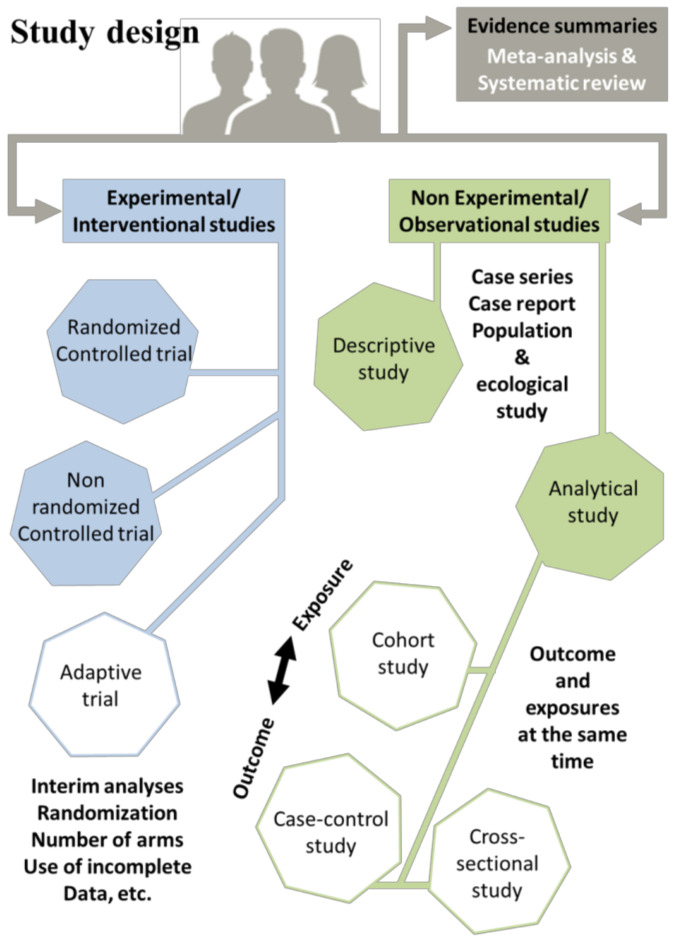
Defining key elements of clinical-research design.

**Figure 4 healthcare-11-02254-f004:**
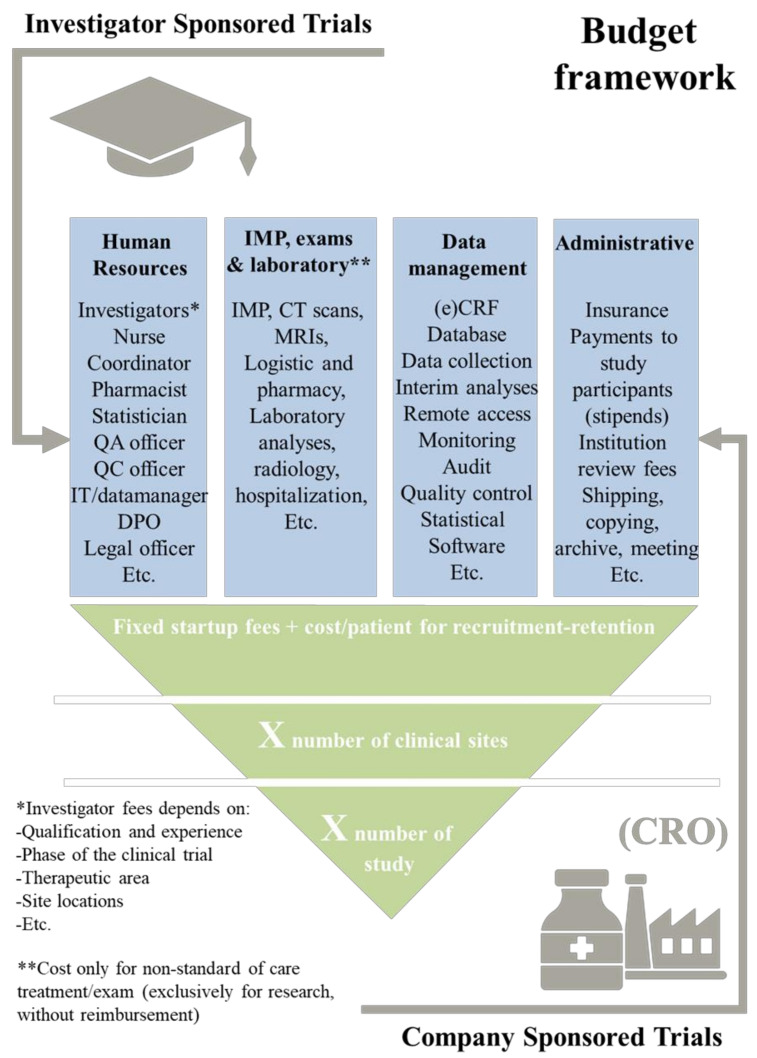
Cost overview of a clinical-research project: financial link between the investigator and company-sponsored trials.

**Figure 5 healthcare-11-02254-f005:**
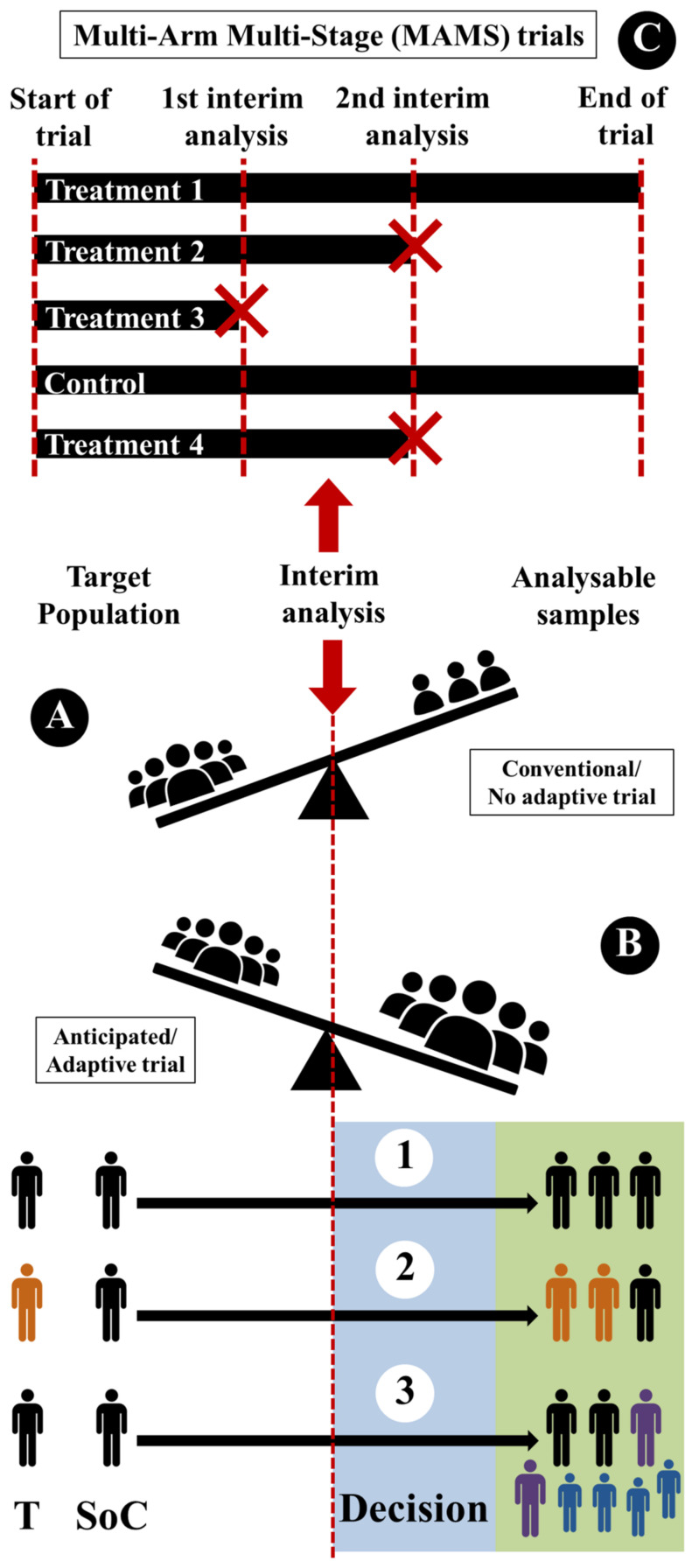
Use of adaptive (**B**,**C**) compared to conventional trials (**A**) in order to prevent the main discrepancies observed after interim analysis, like sample-size re-assessment (**B1**), drug efficacy (**B2**), and criteria for subject selection (**B3**). Multi-arm multi-stage (MAMS) trials are good examples of drug-efficacy selection (**C**).

**Figure 6 healthcare-11-02254-f006:**
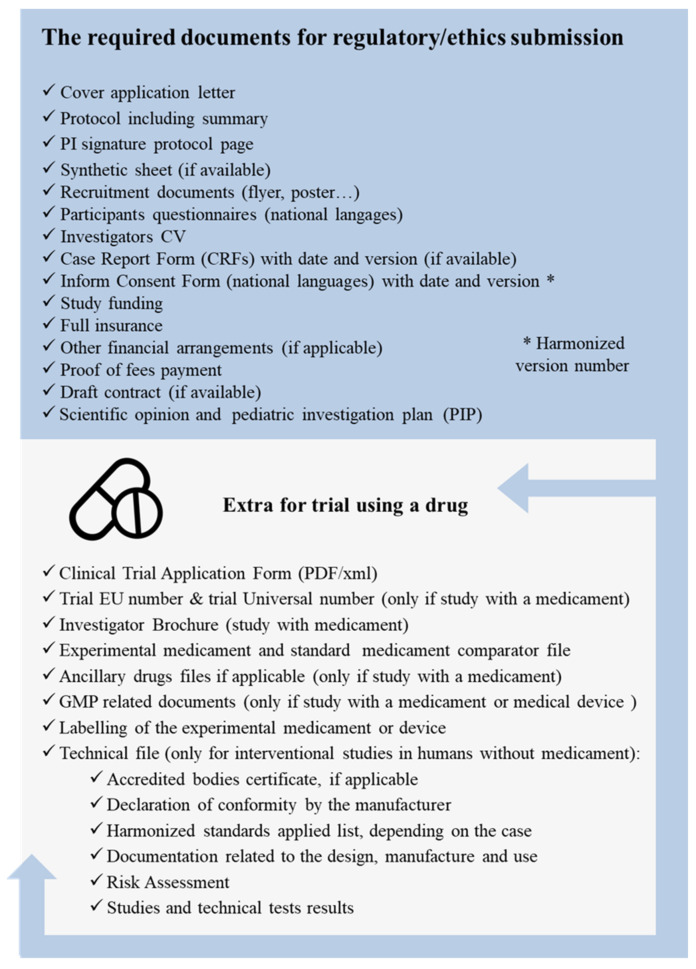
Overview of the required documents for regulatory/ethics submission (additional documents are required for trials using drugs).

**Table 1 healthcare-11-02254-t001:** Ideal research teams. The PI must ensure that each of the functions (sometimes with special areas of expertise, like IT or statistical analyses) described in Table 1 is carried out. Regardless of size, all research teams (generally employed by the research center) perform similar roles and, in most cases, one person carries out several of these activities.

Role in Clinical Research	Responsibility
Principal investigator (PI)	Responsible for the entire study (funding, design, data, staffing, conduct, and reporting data and outcomes through publications.
Additional investigator (co-PI)	Shares responsibility with the PI for ensuring that study is conducted in compliance with applicable laws and regulations and institutional policy.
Project manager	Management of all study activities.
Research assistant (nurse or clinical research associate)	In contact with the patients, ensures that the recruitment (according to the eligibility criteria) is smooth and that the planned visits are carried out.
Quality-control coordinator	Oversees quality control and verifies that all staff are following standard operating procedures (SOPs).
IT manager	Coordinates the organization, implementation, and management of IT processes.
Data manager	Implements the data-entry system and management.
Programmer/analyst	Data-quality analyses.
Statistician	Vital role in the design (including sample size), planning, and analysis of clinical trials (including safety-monitoring guidelines).
Administrative assistant	Ensures the smooth operation and organization of various administrative tasks throughout the trial process.
Legal officer	Coordinates legal contracts with collaborators and sponsors.
Data-protection officer	Verifies that personal data used in trials are in accordance with local data-protection laws.
Financial manager	Verifies specific research budget and manages invoices.
Human-resources manager	Handles the human resources necessary to conduct research.

**Table 2 healthcare-11-02254-t002:** The contents of a clinical-research protocol, including the following topics: A. general information, B. background, C. purpose and objectives, D. methodology and design, E. selection and withdrawal criteria, F. investigational product treatment, G. efficacy, H. safety, I. statistics, J. access to source data/documents, K. quality control and quality assurance, L. ethics and regulatory committees, M. data handling and record keeping, N. financing and insurance, O. publication policy, P. supplements.

Contents	Specificity	Topics
Title, identifying number (EudraCT if applicable), version, and date	Must be modified in case of amendment (version/date)	A
Sponsor details	Monitor (if different from the sponsor for, e.g., CRO)	
Contact details of the person(s) authorized by the sponsor to validate/sign the protocol	Even way for amendments	
Details of the on-site PI who is responsible for conducting the study	Include the full contact details for the trial site(s)	
Details of all medical departments/institutions involved in the trial	Include subcontracted laboratories/platform	
Full description of the investigational medical product (IMP)	Include drug register	B
A summary of previous results from clinical and non-clinical studies that are relevant for the trial	References to relevant research and data	
A summary of the potential risks and benefits to human subjects (if any)	Include all previous preclinical studies	
Scientific justification of dosage, route of administration, and treatment period(s) of IMP		
Legal statement that the trial will be conducted in compliance with the GCP, validated protocol version, and the applicable data-protection requirement(s)	Include GDPR requirement(s)	
Description of the population evaluated		
Description of the purpose and objectives of research		C
Description of the primary endpoints and the secondary endpoints	Include sample-size calculation	
A description of the type/design and procedures	(e.g., placebo-controlled, double-blind, crossover design, etc.)	
Explanation of the methodology followed to minimize risk and avoid bias	(e.g., blinding and randomization processes)	
Description of the management of IMP	Include description of the dosage form, packaging, and labeling	D
Expected duration of subject participation	Duration and number of follow-ups	
Description of the study-discontinuation criteria		
IMP-accountability procedures		
Description of IMP-randomization codes	(e.g., breaking codes in case of emergency)	
Definition of source data to be recorded on the CRFs	(electronic or paper records of data)	
Subject-inclusion/exclusion criteria		
Subject-withdrawal criteria (when and how)	Follow-up and how withdrawn subjects will be replaced	E
The description of IMP used in the trial	Investigator’s brochure included	F
Medication(s) permitted/not permitted before and/or during the trial	Include rescue medication	
Monitoring of subject compliance		
Efficacy-parameter evaluation	Assessment, recording, and analysis of efficacy parameters	G
Safety-parameter evaluation	Assessment, recording, and analysis of safety parameters	
Procedures for recording and reporting adverse events	Follow-up of adverse events.	H
Description of the statistical methods to be employed	Planned interim analysis	
The estimated number of participants to be enrolled (total target number)	Sample size, including power calculations (should be specified in case of multicenter trials)	
Trial-termination criteria		
Accounting procedure for unused and missing clinical data.		
Deviation(s)-reporting procedure	From the original statistical plan, should be justified in protocol	I
Subject-selection criteria	(e.g., eligibility, dosage, and randomization criteria)	
Direct access to source data/document statement	Sponsor and IRB/IEC/regulatory monitoring	J
The quality plan	Describes quality-assurance practices and processes	K
Description of ethical and regulatory considerations relating to the trial		L
Description of the following trial-related procedures: data handling, data verification, statistical analyses, and preparation of trial reports		M
Financing of the project and insurance for participant details	Can be covered in a separate document	N
Publication policy between the partners	Can be covered in a separate document	O
Supplements		P

## Data Availability

Not applicable.

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
