# Peer review of "Design a Clinical Research Protocol: Influence of Real-World Setting"

_healthcare, 2023, doi:10.3390/healthcare11162254_

Round 1
Reviewer 1 Report
The current manuscript proposes a guide to creating and structuring a research protocol. Beyond this is not a new research topic, the manuscript lacks a structure and sequence that allows the reader to clearly understand the steps to follow in designing a clinical study.
Being the title of the manuscript is "Design a clinical research protocol: influence of real-world setting", rather than describing the process of designing a study protocol in general, I suggest that the authors focus on the benefits and challenges associated with designing and conducting real-world evidence studies, highlighting the aspects that are different from traditional clinical studies.
Author Response
Reviewer n°1
The current manuscript proposes a guide to creating and structuring a research protocol. Beyond this is not a new research topic, the manuscript lacks a structure and sequence that allows the reader to clearly understand the steps to follow in designing a clinical study. Being the title of the manuscript is "Design a clinical research protocol: influence of real-world setting", rather than describing the process of designing a study protocol in general, I suggest that the authors focus on the benefits and challenges associated with designing and conducting real-world evidence studies, highlighting the aspects that are different from traditional clinical studies.
We would like to thank sincerely the reviewer 1 for their thoughtful comments and efforts towards improving our manuscript.
There is a narrative review (87 references), a comprehensive methodological guide (as complete as possible) which presents an approach to design a research protocol, offering a detailed step-by-step process for doctors and researcher. Additionally, it provides an insightful review of prominent standardized clinical research guidelines that emerging researchers should anticipate encountering as they formulate their protocols. So as narrative review there is definitively not results but a discussion with a selection of solutions, opportunities and perspectives based on the literature to improve protocol research design.
Regarding structure:
The introduction is meant to be the most precise and capable of giving the readers context by including pertinent references.
The methodology has been covered in great length with all these sub-points:
2.1 Building blocks of clinical research protocol: Eight essential blocks that progress incrementally, starting from the conceptual stage prior to drafting, advancing to the initial drafting phase, then moving towards submission and regulatory validation, ultimately culminating in the publication of the protocol.
2.2. Before writing phase: Feasibility, study aims and methodology the most common high designs included like Case-Control Study, Cohort Study, and Cross-sectional study reframed in one paragraph as it seems more like a book chapter.
2.3. Writing phase: the major guidelines and observed discrepancies in clinical research
2.4. How internal review by study team is important.
2.5. Hot topic regarding regulatory and ethics validations
2.6. Mandated registration of the protocol
- As suggested, the benefits and challenges associated with designing and conducting real-world evidence studies has been included in the discussion (see the modifications of the manuscript in track-change):
“Another critical point is not sufficiently taking into account the utilization of real-world evidence (RWE) clinical studies in protocol design. RWE clinical studies have emerged as a valuable approach in medical research, offering several benefits and challenges. These studies provide enhanced external validity by reflecting real-world patient populations more accurately compared to traditional randomized controlled trials (RCTs). RWE studies encompass a broader spectrum of patients, leading to broader generalizability and informed clinical decision-making based on individual patient needs. Additionally, RWE studies offer longitudinal insights by observing patients over extended periods, revealing long-term treatment outcomes and safety profiles. One of the significant advantages of RWE studies is their cost-effectiveness. Leveraging existing data sources like electronic health records and claims databases reduces the need for extensive infrastructure and patient recruitment, expediting the evidence generation process. Ethically, RWE studies address certain concerns by observing treatment effects without intervening in patient care, avoiding the need for placebo-controlled arms in some cases.
However, RWE studies also face several challenges. Ensuring data quality and completeness is paramount, as electronic health records and claims databases may contain missing or erroneous information, potentially introducing bias. Selection bias and confounding factors are inherent risks in RWE studies, as patients may self-select treatments, leading to imbalances in patient characteristics across treatment groups. Statistical techniques like propensity score matching and sensitivity analyses are employed to mitigate these biases, but their effectiveness relies on the quality and breadth of available data. The lack of standardization in RWE studies presents another challenge, as protocols and data collection practices can vary significantly across studies, making direct comparisons difficult and hindering generalizability. Data privacy and security concerns arise with the increasing reliance on electronic health records and digital sources, necessitating robust data anonymization and secure data sharing protocols to protect patient confidentiality and comply with data protection regulations. Establishing causality in RWE studies is challenging due to their observational nature. Unlike RCTs with randomization and control groups, RWE studies rely on statistical methodologies and causal inference techniques to establish links between treatments and outcomes. While sophisticated analytical approaches exist, they cannot fully account for all potential confounders, leading to inherent uncertainties in causal interpretations [83, 84].”
Reviewer 2 Report
The current review, “Design a clinical research protocol: influence of real-world setting,” This comprehensive guide presents a systematic approach to design a research protocol, offering a detailed step-by-step process. Additionally, it provides an insightful review of prominent standardized clinical research guidelines that emerging researchers should anticipate encountering as they formulate their protocols. The study can be interesting and helpful to the doctors. However, before this work can be accepted, some concerns must be addressed by emphasizing the following points.
Issues:
1. The introduction seems to be precise and capable of giving the readers context by including pertinent references. However, some point’s needs to be corrected for e.g. could you please provide the reference for the data source used to authorize research projects approved annually by the European Union? (Line 24)
2. In the prospective study different full form such as EU, SPIRIT, ICH-GCP, CONSORT, etc. are not provided, kindly add them.
3. The methodology has been covered in great length and all the sub-points have been covered in great length.
a. In subsection 2.1 Building blocks of clinical research protocol there are some typological error which needs to be corrected for e.g. in line 117 ‘reference 23’ and ‘references 24-28’ needs to be corrected.
b. In subsection 2.2. Before writing phase: Feasibility, study aims and methodology the most common high designs included like Case-Control Study, Cohort Study, and Cross-sectional study should be reframed in one paragraph as it seems more like a book chapter.
4. The discussion and conclusion is clearly explained, providing proper insight of the study.
5. Some of the references are too old, that needs to be updated. For e.g. reference 56 used for Cohort Study; and other references including 37, 38, 54 etc.
The review article exhibiting a well-planned structure that effectively addressed all the essential aspects. The content was thoroughly presented, offering a comprehensive analysis of the subject matter. However, during the review process, some minor errors were identified that require attention in order to enhance the overall accuracy and clarity of the article.
Author Response
The current review, “Design a clinical research protocol: influence of real-world setting,” This comprehensive guide presents a systematic approach to design a research protocol, offering a detailed step-by-step process. Additionally, it provides an insightful review of prominent standardized clinical research guidelines that emerging researchers should anticipate encountering as they formulate their protocols. The study can be interesting and helpful to the doctors. However, before this work can be accepted, some concerns must be addressed by emphasizing the following points. The review article exhibiting a well-planned structure that effectively addressed all the essential aspects. The content was thoroughly presented, offering a comprehensive analysis of the subject matter. However, during the review process, some minor errors were identified that require attention in order to enhance the overall accuracy and clarity of the article.
Author response
We would like to thank sincerely the reviewer 2 for his positive enthusiasm and for their thoughtful comments and efforts towards improving our manuscript. The reviewer 2 understood that this comprehensive guide presents a systematic approach to design a research protocol, offering a detailed step-by-step process. R2 will find our answers/modifications directly in the comments.
Issues:
1.The introduction seems to be precise and capable of giving the readers context by including pertinent references. However, some point’s needs to be corrected for e.g. could you please provide the reference for the data source used to authorize research projects approved annually by the European Union? (Line 24).
Author response
Thank you so much for this comment. Indeed, there was an topographic error! it is indeed 4,000 and not 24,000 clinical trials authorized per year in the EU. As suggested by you, the reference “Kohnstamm, N. M. (2019). Approaching Judgment Day: The Influence of Brexit on the EU Pharmaceutical Framework. Legal Issues of Economic Integration, 46(2).” has been included, see the modifications of the manuscript in track-change).
- In the prospective study different full form such as EU, SPIRIT, ICH-GCP, CONSORT, etc. are not provided, kindly add them.
Author response
Thank you for your comments. As suggested by you, the full form of EU, SPIRIT, ICH-GCP, CONSORT has been included, see the modifications of the manuscript in track-change).
3.The methodology has been covered in great length and all the sub-points have been covered in great length.
- In subsection 2.1 Building blocks of clinical research protocol there are some typological error which needs to be corrected for e.g. in line 117 ‘reference 23’ and ‘references 24-28’ needs to be corrected.
- In subsection 2.2. Before writing phase: Feasibility, study aims and methodology the most common high designs included like Case-Control Study, Cohort Study, and Cross-sectional study should be reframed in one paragraph as it seems more like a book chapter.
Author response
Thank you so much for this comment. The typological errors has been changed, see the modifications of the manuscript in track-change). As suggested by you, the most common high designs included like Case-Control Study, Cohort Study, and Cross-sectional study should be reframed in one paragraph as it seems more like a book chapter, see the modifications of the manuscript in track-change).
- The discussion and conclusion is clearly explained, providing proper insight of the study.
Thank you so much for this positive comment.
- Some of the references are too old, that needs to be updated. For e.g. reference 56 used for Cohort Study; and other references including 37, 38, 54 etc.
Thank you so much for this positive comment.
New reference 56 : Camargo, L. M. A., Silva, R. P. M., & de Oliveira Meneguetti, D. U. (2019). Research methodology topics: Cohort studies or prospective and retrospective cohort studies. Journal of Human Growth and Development, 29(3), 433-436.
New reference 37 : Elliott, M. R. (2023). Surrogate Endpoints in Clinical Trials. Annual Review of Statistics and its Application, 10, 75-96.
New reference 38 : Walia, A., Haslam, A., & Prasad, V. (2022). FDA validation of surrogate endpoints in oncology: 2005–2022. Journal of Cancer Policy, 34, 100364.
New reference 54 : Setia, M. S. (2016). Methodology series module 2: case-control studies. Indian journal of dermatology, 61(2), 146.
We would like to thank sincerely the reviewer 2 for his positive enthusiasm and for their thoughtful comments and efforts towards improving our manuscript.
Reviewer 3 Report
I reviewed this scientific study; however, it did not correspond to the current section, the review article. If there is one, it should be covered within the study protocol. The goal, as indicated in the introduction, is to provide a comprehensive guideline. Overall, the review does not adhere to the standard format. The authors did not specify how the literature was gathered and assessed. There was also no result section, indicating that the current structure is for study protocol. The discussion is very brief for a review article, and the methods appear to be too detailed to grasp. The authors should keep the method short and emphasize only the gaps or any unusual actions required to complete the process. Otherwise, it appears to be a standard lecture note with no emphasis on the additional knowledge provided.
Author Response
Reviewer n°4
I reviewed this scientific study; however, it did not correspond to the current section, the review article. If there is one, it should be covered within the study protocol. The goal, as indicated in the introduction, is to provide a comprehensive guideline. Overall, the review does not adhere to the standard format. The authors did not specify how the literature was gathered and assessed. There was also no result section, indicating that the current structure is for study protocol. The discussion is very brief for a review article, and the methods appear to be too detailed to grasp. The authors should keep the method short and emphasize only the gaps or any unusual actions required to complete the process. Otherwise, it appears to be a standard lecture note with no emphasis on the additional knowledge provided.
We would like to thank sincerely the reviewer 4 for their thoughtful comments and efforts towards improving our manuscript.
There is a narrative review (87 references), a comprehensive methodological guide (as complete as possible) which presents an approach to design a research protocol, offering a detailed step-by-step process for doctors and researcher. Additionally, it provides an insightful review of prominent standardized clinical research guidelines that emerging researchers should anticipate encountering as they formulate their protocols. So as narrative review there is definitively not results but a discussion with a selection of solutions, opportunities and perspectives based on the literature to improve protocol research design.
Regarding structure:
The introduction is meant to be the most precise and capable of giving the readers context by including pertinent references.
The methodology has been covered in great length with all these sub-points:
2.1 Building blocks of clinical research protocol: Eight essential blocks that progress incrementally, starting from the conceptual stage prior to drafting, advancing to the initial drafting phase, then moving towards submission and regulatory validation, ultimately culminating in the publication of the protocol.
2.2. Before writing phase: Feasibility, study aims and methodology the most common high designs included like Case-Control Study, Cohort Study, and Cross-sectional study reframed in one paragraph as it seems more like a book chapter.
2.3. Writing phase: the major guidelines and observed discrepancies in clinical research
2.4. How internal review by study team is important.
2.5. Hot topic regarding regulatory and ethics validations
2.6. Mandated registration of the protocol
- As suggested by you and another reviewer, the discussion has been lengthened by the addition of a paragraph regarding the benefits and challenges associated with designing and conducting real-world evidence studies has been included in the discussion (see the modifications of the manuscript in track-change): “Another critical point is not sufficiently taking into account the utilization of real-world evidence (RWE) clinical studies in protocol design. RWE clinical studies have emerged as a valuable approach in medical research, offering several benefits and challenges. These studies provide enhanced external validity by reflecting real-world patient populations more accurately compared to traditional randomized controlled trials (RCTs). RWE studies encompass a broader spectrum of patients, leading to broader generalizability and informed clinical decision-making based on individual patient needs. Additionally, RWE studies offer longitudinal insights by observing patients over extended periods, revealing long-term treatment outcomes and safety profiles. One of the significant advantages of RWE studies is their cost-effectiveness. Leveraging existing data sources like electronic health records and claims databases reduces the need for extensive infrastructure and patient recruitment, expediting the evidence generation process. Ethically, RWE studies address certain concerns by observing treatment effects without intervening in patient care, avoiding the need for placebo-controlled arms in some cases.
However, RWE studies also face several challenges. Ensuring data quality and completeness is paramount, as electronic health records and claims databases may contain missing or erroneous information, potentially introducing bias. Selection bias and confounding factors are inherent risks in RWE studies, as patients may self-select treatments, leading to imbalances in patient characteristics across treatment groups. Statistical techniques like propensity score matching and sensitivity analyses are employed to mitigate these biases, but their effectiveness relies on the quality and breadth of available data. The lack of standardization in RWE studies presents another challenge, as protocols and data collection practices can vary significantly across studies, making direct comparisons difficult and hindering generalizability. Data privacy and security concerns arise with the increasing reliance on electronic health records and digital sources, necessitating robust data anonymization and secure data sharing protocols to protect patient confidentiality and comply with data protection regulations. Establishing causality in RWE studies is challenging due to their observational nature. Unlike RCTs with randomization and control groups, RWE studies rely on statistical methodologies and causal inference techniques to establish links between treatments and outcomes. While sophisticated analytical approaches exist, they cannot fully account for all potential confounders, leading to inherent uncertainties in causal interpretations [83, 84].”
Round 2
Reviewer 3 Report
I reviewed the amended version of the manuscript and found it exciting and improved. Thank you for taking the time to consider the suggestion and respond to the comments. A well done job